# Guillain-Barré syndrome after the Zika epidemic in Colombia: A multicenter, matched case-control study

Lyda Osorio[1‡], Beatriz Parra[2‡], Martha Moyano[1], Reydmar Lopez-Gonzalez[3], Jorge A. Jimenez-Arango[4], José Vargas-Manotas[5,6], Jairo Lizarazo[7], Gustavo E. Ramos-Burbano[8,9], Mario Daniel Llanos[9], Fernando Rosso[10,11], Jonathan Urrego[8,12], Juan P. Rojas[13], Christian A. Rojas[8,12,14], Julie Benavides-Melo[15], Viviana A. Martinez-Villota[16], Karina A. Luque-Burgos[17], Adriana M. Ruiz[17], Liliana Soto[2], Laura Quintero-Corzo[2], Jaime A. Quintero[2], Daniela Zuluaga-Lotero[2], David Acero-Garces[2], Susana C. Dominguez-Peñuela[1,19], Susan Halstead[18], Hugh J. Willison[18], Carlos A. Pardo[19]*, On behalf of the Neuroinfections Emerging in the Americas Study (NEAS)[¶]

1 School of Public Health, Universidad del Valle, Cali, Colombia, 2 Department of Microbiology, Universidad del Valle, Cali, Colombia, 3 Hospital San Vicente Fundación Rionegro, Universidad de Antioquia, Medellín, Colombia, 4 Hospital Alma Máter de Antioquia, Universidad de Antioquia, Medellín, Colombia, 5 Universidad Simón Bolívar, Barranquilla, Colombia, 6 La Misericordia Clínica Internacional, Barranquilla, Colombia, 7 Hospital Universitario Erasmo Meoz, Universidad de Pamplona, Pamplona, Colombia, 8 School of Medicine, Universidad del Valle, Cali, Colombia, 9 Clínica Rey David, Cali, Colombia, 10 Department of Internal Medicine, Fundación Valle del Lili, Cali, Colombia, 11 Universidad Icesi, Cali, Colombia, 12 Hospital Universitario del Valle, Cali, Colombia, 13 Department of Pediatrics, Fundación Clínica Infantil Club Noel, Cali, Colombia, 14 Department of Pediatrics, Clinica Imbanaco, Cali, Colombia, 15 Universidad Cooperativa de Colombia, Pasto, Colombia, 16 Hospital Universitario Departamental de Nariño, Pasto, Colombia, 17 Hospital Universitario Erasmo Meoz, Cúcuta, Colombia, 18 School of Infection and Immunity, University of Glasgow, Glasgow, United Kingdom, 19 Departments of Neurology and Pathology, Johns Hopkins University School of Medicine, Baltimore, Maryland, United States of America

‡ First co-authors
¶ Listed in S1 Appendix
* cpardov1@jhmi.edu

## Abstract

### Background

Zika produced the highest increase in the incidence of Guillain-Barré syndrome (GBS) in Latin America in the last decade. The Neuroinfections Emerging in the Americas Study (NEAS) was established in 2016 to investigate the association of emerging infectious disorders with GBS in Colombia. The present study assessed the role of preceding infections, including arboviruses and other pathogens, as risk factors for GBS.

### Methods

A case-control study was conducted prospectively between June 2016 and December 2019 in 5 Colombian cities. We recruited newly diagnosed patients with GBS and a house control plus an age and season-matched-hospital control per case. Clinical information, blood, CSF, and urine samples were used to diagnose bacterial and viral infections.

**Data availability statement:** The data underlying the results presented in the study are available at the following link: https://osf.io/29VNY/

**Funding:** This work was supported by the National Institutes of Health (NIH) (R01NS110122 to LO, BP and CAP https://www.nih.gov/), the Bart McLean Fund for Neuroimmunology Research (to CAP), the European Union's Horizon 2020 ZikaPLAN Grant Agreement (734584 to LO, BP and CAP https://zikaplan.tghn.org/) and by the Wellcome Trust Grants (092805 and 202789 to HJW https://wellcome.org/). The funders had no role in study design, data collection and analysis, decision to publish, or preparation of the manuscript.

Anti-glycolipid antibodies were identified in serum. Statistical analyses were performed using conditional logistic regression.

## Findings

Fifty-seven patients with GBS, 66·7% male, 52 years of median age, were recruited along with 77 (55 house and 22 hospital) controls. GBS was associated with presenting diarrhea (adjusted OR 10·94; 95% CI 1·8-66·29; p=0·009) and a history of recent upper respiratory tract infection (aOR 13·91; 95% CI 2·38-81·1 p=0·003). Specific recent infections did not significantly differ between cases and controls, but the number of infections was associated with GBS (aOR=1·77 95% CI 1·04-3·01 p=0·03). *C. jejuni* (74%), *M. pneumoniae* (23%), and Chikungunya (7%) were the most frequent infections. Anti-glycolipid IgG against GM1 and their heterodimer complexes were identified to be associated with GBS.

## Conclusions

After the Zika epidemic, infections causing diarrhea and upper respiratory diseases contributed to the burden of GBS in Colombia. Prevention and control of food-borne pathogens could reduce the incidence of GBS in Colombia.

## Author summary

Before the emergence of the Zika virus epidemic in the Americas, the incidence and risk factors associated with GBS were not well documented in Colombia and many Latin American countries. A major outbreak of GBS associated with Zika infection in 2016 and evidence of seasonality and clusters of GBS in some Latin American countries prompted the investigation of the role of infections as risk factors associated with GBS in Colombia. Our study assessed factors associated with GBS in Colombia, which included a wide spectrum of infectious etiologies, bacterial and viral pathogens, and arboviruses recently emerging in the Americas such as Chikungunya and Zika. The evidence further supports that diarrheal disease and upper respiratory infections are the dominant drivers in the pathogenesis of GBS in the post-Zika period in Colombia and possibly in other Latin American countries. *Campylobacter jejuni,* a gastrointestinal pathogen, was the most prevalent infection in GBS, highlighting the significance of diarrheal disease as a risk factor in GBS. The study also underlies the observation that after Zika became endemic in the Americas, particularly in Colombia, the number of cases of Zika-associated GBS decreased and paralleled the lower incidence of Zika in the overall population.

## Introduction

Guillain–Barré syndrome (GBS) is a rapidly progressive immune-mediated polyradiculoneuropathy and the most frequent cause of acute flaccid paralysis worldwide [1]. GBS can be life-threatening and frequently occurs as a post-infectious disease [2], although non-infectious triggers have also been described [3]. The role of preceding infections as a risk factor for GBS has been highlighted lately by emerging viral infectious disorders as major clusters of GBS were observed during the outbreaks of Zika virus (ZIKV) infection in the French Polynesia in 2013-2014 [4], and in Latin American countries [5] including Colombia [6] in 2015 and 2016.

Most recently, outbreaks of GBS in Latin America have also occurred in the context of *Campylobacter jejuni* infection [7]. The present study assessed the role of preceding illnesses and infectious disorders, including arboviruses and other pathogens, as risk factors for GBS.

## Methods

### Ethics statement

Written informed consent was obtained from each study participant or their legally authorized representative. Assent was required for minors above 7 years of age. Consent capacity was determined based on the subject's ability to communicate his or her understanding of the research, risks, and benefits. The appropriate surrogate decision maker was identified if the subject could not provide this information. Consent for children participants was obtained from the parent or guardian. If the participant's neurological condition improved during the conduction of the research activities, consent capacity was reassessed. If the participant regained capacity, the participant was reconsented. Ethical approval was granted by the Ethics Committees of Universidad del Valle and each participating hospital in Colombia (Universidad del Valle, Cali-Colombia. IRB - Project ID 034-016; code 006-016).

### Study design and population

A case-control study was conducted between June 2016 and December 2019 in 9 university-based hospitals across five Colombian cities (Barranquilla, Cali, Cucuta, Medellin, and Pasto), which are part of the Neuroinfections Emerging in the Americas Study (NEAS), a research network established in Colombia as a multicenter-based observatory of acute neuroinflammatory disorders to investigate the role of emerging viral infections in neuroinflammatory diseases. The study was designed to collect data concurrently on cases and controls only once, at the time of enrollment. Cases were all patients with newly diagnosed GBS, including the Miller Fisher syndrome (MFS) variant, admitted to the hospitals who fulfilled the Brighton criteria [8,9]. To minimize selection bias, we included two types of controls per case: a hospital and a house control. These controls were specifically selected to reflect the hypothesized exposure (infections) without the outcome of interest (GBS). Hospital controls were age-matched (age difference +/- 10 years), admitted to the same hospital +/- 15 days of admission of the corresponding case, with a febrile illness or suspected acute viral infection, without or with non-specific neurological symptoms. House controls were required to have resided with the case at least during the 3 months before the case's symptoms onset. House controls were not age-sex matched with the cases. Subjects with a known etiology of the neurological disorder (trauma) or a history of a neuroinflammatory disease (e.g., multiple sclerosis, myelitis) were excluded. Study research coordinators consecutively identified eligible cases and hospital controls through neurology rounds/consults and screening hospital admission logs. House controls were identified by asking the patient or companion to list the persons who live with them. Hospital and house controls were selected using simple random sampling, when possible, otherwise by convenience sampling. The sample size was calculated as 50 per group to identify an odds ratio-OR=4 (e.g., 50% of cases and 20% of controls with evidence of infection) with a 5% significance level and a statistical power of 90%.

### Clinical data and sample collection

Demographic and clinical information that included general and neurological symptoms and signs, past medical history, laboratory, imaging, and ancillary testing information was collected by interview and examination performed by the study clinicians and/or retrieved from electronic medical records using a standardized approach [6]. Neurological disability

was classified according to the GBS Disability Score [10], modified Rankin scale (mRS) [11], and Medical Research Council-MRC Sum Score [12] to evaluate global muscle strength. Neurophysiological data were collected from GBS cases and classified into electrodiagnostic types using the Hadden et al. criteria [13]. Data were collected locally and transferred by the study coordinators into Research Electronic Data Capture-REDCap tools hosted at the Johns Hopkins University.

## Laboratory procedures and infection assessment

Biological samples, including venous blood, urine, and cerebrospinal fluid (CSF) were collected as close as possible to the enrollment date. Biological samples were kept refrigerated (2–8°C) or frozen (-80°C) and transported to the central laboratory at Universidad del Valle, Cali, Colombia, where they were aliquoted and stored at -80°C until they were processed. Infection assessment included a combination of molecular and immunoserological studies of biological samples. Samples were thawed in batches at room temperature and maintained at 4°C before immunological or molecular testing was performed at the NEAS laboratory at the Department of Microbiology of the Universidad del Valle. The diagnosis of ZIKV, and Chikungunya virus (CHIKV) infection was based on real-time quantitative Reverse Transcription Polymerase Chain Reaction (RT-PCR) [14–16] in serum, urine, and CSF. The real-time RT-PCR for ZIKV was considered positive when both target genes were positive with a threshold cycle-CT <=38 together with a typical sigmoid PCR amplification curve, negative when both genes were undetectable (CT>38), and equivocal when only 1 of the target genes was positive [14]. The real-time RT-PCR for CHIKV was considered positive when the target gene showed a CT <=38 with a typical sigmoid PCR amplification curve and negative when the gene was undetectable [16]. Diagnosing dengue virus (DENV) infection used a nested RT-PCR for DENV [17,18] in serum and CSF. The test was considered positive when a typical deoxyribonucleic acid-DNA electrophoretic band for any viral serotypes was observed and negative if no DNA bands were observed. Immunoassay testing of serum for DENV used Enzyme Linked Immunosorbent Assay-ELISA techniques targeting anti-DENV Immunoglobulins M (IgM) and G (IgG) (cat 01PE20 and cat 01PE10, PANBIO, Alere Inc., Waltham, MA). The anti-DENV was interpreted as anti-flavivirus because of cross-reactivity between ZIKV and DENV antibodies [14,19]. The serologic diagnosis of flavivirus infection was interpreted as recent infection when IgM was positive with IgG either positive or negative, and past flavivirus infection (here called exposed) when IgM was negative and IgG was positive [20], or negative when IgM and IgG were both negative.

The immunoassay diagnosis of *C. jejuni* infection or previous exposure was based on testing of anti-*C. jejuni* IgM (cat. ESR139M, SERION diagnostics, Würzburg, Germany), IgA and IgG (cat. EI 2091-9601 A and EI 2091-9601 G, EUROIMMUN, Lübeck, Germany). *C. jejuni* infection was considered recent when two or three immunoglobulin isotypes were positive, previous exposure when only one immunoglobulin was positive or negative when the three immunoglobulins were negative [21–23]. Assessment of *M. pneumonie* and hepatitis E virus (HEV) infection was based on testing of anti-*M. pneumoniae* IgM (cat. EI 2202-9601 M EUROIMMUN, Lübeck, Germany), anti-HEV IgM (cat. EI 2525-9601 M, EUROIMMUN, Lübeck, Germany); For HEV and *M. pneumoniae* a positive IgM was interpreted as a recent infection, and a negative IgM ruled out infection [24]. Testing of CMV infection used anti-CMV IgM (cat. EI 2570-9601 M, EUROIMMUN, Lübeck, Germany) followed by IgG and IgG avidity test (cat. EI 2570-9601-1 G, EUROIMMUN, Lübeck, Germany) when IgM was positive or equivocal. Similarly, diagnosis of VZV infection used anti-VZV IgM (cat. EI 2650-9601 M, EUROIMMUN, Lübeck, Germany) followed by IgG and IgG avidity test when IgM was positive or equivocal (cat. EI 2650-9601-1 G, EUROIMMUN, Lübeck, Germany). For CMV and

VZV, a negative IgM ruled out the diagnosis; if IgM was positive or equivocal and IgG avidity was low (IgG optical density-OD value reduction below 40% after urea treatment), it was considered a primary infection, if it was high (IgG OD value above 60% after urea treatment), it was considered as reactivation [22,25]. Finally, testing for Epstein-Barr virus (EBV) used anti-EBV viral capsid antigen-VCA IgM antibodies (cat. EI 2791-9601 M, EUROIMMUN, Lübeck, Germany) followed by anti-Epstein-Barr nuclear antigen-EBNA IgG (cat. EI 2793-9601-G, EUROIMMUN, Lübeck, Germany) when anti-VCA was positive or equivocal. A positive anti-EBV VCA IgM, and a negative anti-EBNA IgG test were considered a primary infection, and both positive as reactivation. If IgM was negative, infection was ruled out [22,26]. Results that were classified as indeterminate or equivocal were repeated once. For quality assurance, the OD and index values for the positive, negative, and calibrator controls were within the range values defined for each lot, according to the manufacturers. RT-PCR plate assays always included positive and negative samples. The laboratory microbiologists were blind to the samples' case/control diagnosis status.

## Anti-glycolipid testing

Serum samples were screened for the presence of specific anti-glycolipid IgG, IgM and IgA antibodies in GBS cases and controls using a multiplexed array panel including 16 single glycolipids (GM1, GM2, phosphatidylserine [PS], GM4, GA1, GD1a, GD1b, GT1a, GT1b, GQ1b, GD3, SGPG, LM1, GalCNAc-GD1a, GalC, and sulfatide), and 120 heteromeric 1:1 (v:v) complexes. Laboratory testing was performed at the Glasgow Biomedical Research Centre, University of Glasgow, Glasgow, UK, using protocols previously published [27]. Antibody-antigen binding was detected using fluorescent-conjugated anti-human immunoglobulin specific secondary antibodies, with fluorescent intensity being measured on a scale of 0–65,535 using a Genepix 4300A (Molecular Devices, San Jose, CA) microarray scanner. Antibody intensity values were reported as the average of duplicate median fluorescent intensity values per sample [7].

## Statistical analysis

Descriptive analyses were performed for demographic, clinical and laboratory data separately for matched and unmatched GBS cases, hospital, and house controls. Categorical variables were presented as frequencies and quantitative variables with means, standard deviations, or medians and ranges as appropriate. To estimate the number of infections per person, we add up positive results obtained for each microorganism tested (i.e., evidence of infection with one or more microorganisms). For the analyses of Anti-glycolipid antibodies, we identified cut-off values using the 95th percentile of the results obtained in house controls. Heatmaps of anti-glycolipid IgGs were developed in R Studio 2023 06.0+421, using the gplots 3.1.3 and ComplexHeatmap 2.10.0 packages, based on hierarchical clustering of Pearson's correlation distant method [28]. Bivariate and multiple conditional logistic regression models were fitted for GBS cases and their corresponding control pairs, and in a subgroup of *C. jejuni* positives, to obtain crude and adjusted ORs with 95% CI. Hospital and house controls were pooled together to increase the precision of estimates, and the source of controls was considered in all models. The backward approach was used for modeling, and the Likelihood Ratio-LR test and Akaike-information criteria were used for model comparison. A two-sided p-value of <0.05 was considered statistically significant, except during bivariate analysis of anti-glycolipids, where the Bonferroni correction for multiple testing was considered. Analyses were performed using STATA version 14.0 for Windows (StataCorp. 2015. Stata Statistical Software: Release 14. College Station, TX: StataCorp LP).

## Results

### Study population

A total of 82 patients with GBS and 77 control subjects were enrolled. Fifty-seven GBS cases were matched with at least one control of whom 20 GBS cases were matched with both house and hospital controls, 35 GBS with only house controls, and 2 GBS with only hospital controls. Twenty-five cases of GBS did not have a matched control (Fig 1). The clinical, demographic, and microbiological characteristics of unmatched GBS patients were similar to those of matched cases included in the analysis, except that the latter showed a higher frequency of primary axonal phenotype, and of *C. jejuni* and *M. pneumoniae* recent infections (S1 and S2 Tables). The median age of the GBS cases was 52 years (range 3–82 years), of the hospital controls was 60 (6–79) years, and of the house controls was 45 (11–77) years. GBS cases were predominantly males (67%) in contrast to both control groups where males were 27% in the hospital group, and 22% in the house group. The most frequent systemic presenting symptoms of the GBS cases included asthenia, fever, diarrhea, myalgias, and headaches; however, 25% of them did not have preceding systemic symptoms. These symptoms were also present in the hospital controls, most frequently fever (96%) and asthenia (73%) (Table 1).

Clinical, CSF, and neurophysiological features and treatments for GBS subjects are described in Table 2. The median time from initial symptoms to the start of neurological symptoms were 5 (range 0–35) days for the GBS cases. The most frequent neurological symptoms were motor dysfunction, including ascending paralysis (58%), lower limb paralysis (75%), upper limb paralysis (49%), and sensory symptoms (54%). The frequencies of facial paralysis and dysautonomia were 9% each. Early strength measures of GBS cases had a median MRC Sum Score of 30 at nadir. At the same time, mRS established a moderately severe and severe disability score (mRS $\geq$ 4) at nadir in 81% of the GBS subjects. The GBS disability

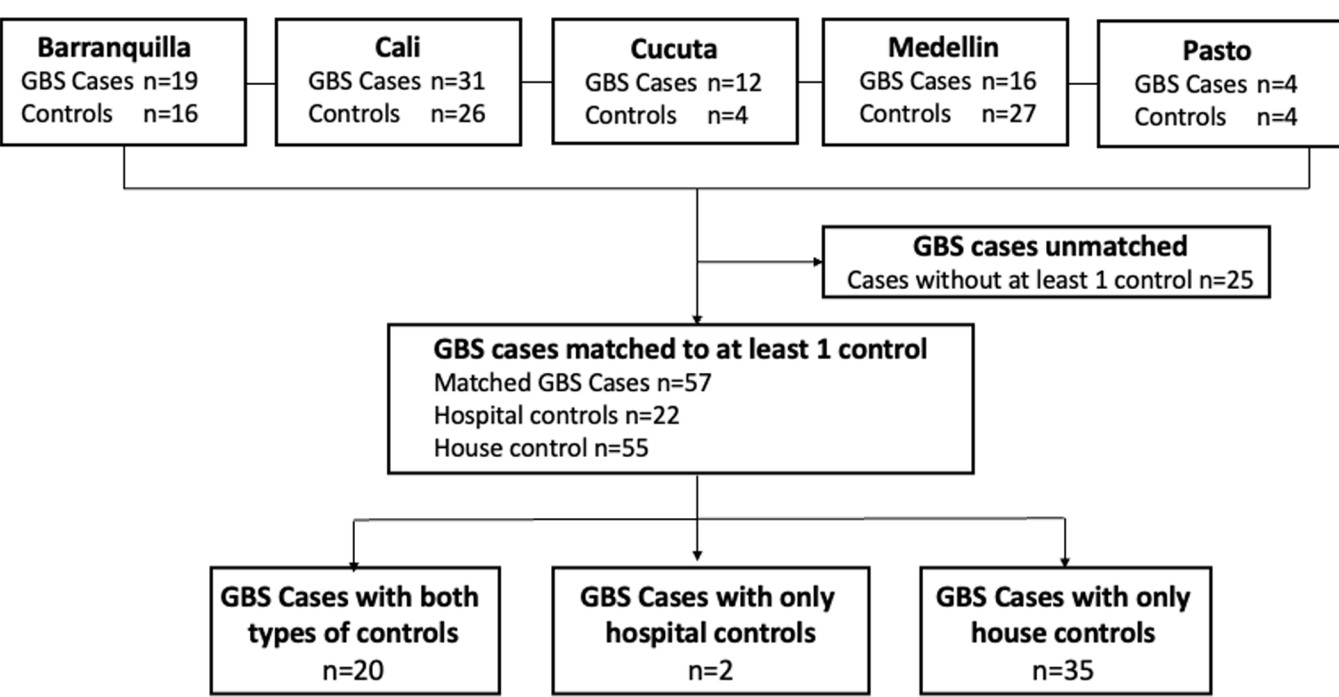

**Fig 1. Flow diagram of the selection process of GBS cases and controls across study locations.**

**Table 1.** Demographics and presenting symptoms[a] of GBS cases and controls.

| Characteristics | GBS cases N=57 (%) | Hospital controls N=22 (%) | House controls N=55 (%) |
|---|---|---|---|
| Age in years, Median (Range) | 52 (3–82) | 60 (6–79) | 45 (11–77) |
| Age >15 years old | 50 (88) | 20 (91) | 53 (98) |
| Sex Male | 38 (67) | 6 (27) | 12 (22) |
| Upper respiratory infection or influenza | 19 (33) | 3 (14) | 2 (4) |
| Diarrhea at onset | 17 (30) | 0 | 1 (2) |
| Urinary tract infection | 0 | 2 (9) | 1 (2) |
| Vaccination | 0 | 0 | 1 (2) |
| Asthenia | 28 (49) | 16 (73) | 2 (4) |
| Fever | 20 (35) | 21 (96) | 3 (6) |
| Preceding diarrhea no at onset (within 4 weeks) | 16 (28) | 6 (27) | 1 (2) |
| Myalgias | 16 (28) | 8 (36) | 3 (6) |
| Nausea and/or vomiting | 11 (19) | 6 (27) | 0 |
| Headache | 11 (19) | 7 (32) | 2 (4) |
| Arthralgia | 7 (12) | 7 (32) | 2 (4) |
| Rash | 3 (5) | 5 (23) | 0 |
| Asymptomatic [b] | 14 (25) | 0 | 51 (93) |

[a]Symptoms of systemic illness immediately preceding or during the onset of neurological symptoms.

[b]No evidence of systemic illness when neurological symptoms presented

score indicated a poor prognosis (score ≥ 3) in 93% of participants. The most frequent electro-physiological phenotype was primary demyelinating (42%) followed by primary axonal (35%). These two electrophysiological phenotypes are closely concordant with the motor-sensory (demyelinating) and pure motor (primary axonal) clinical phenotypes. [29,30]The most common treatment for GBS subjects was IVIG (54%), while plasma exchange was used in 35%. Children <15 years old were more frequently admitted to the ICU (Table 2). Information about biological sample collection related to time of treatment is described in Table S1.

## Preceding clinical history and infection assessment

The laboratory testing and assessment of viral and bacterial infections in the GBS cases and controls are summarized in Table 3 and S2 Table. In the GBS subjects, the most common serological evidence of recent infections was by *C. jejuni* (33%), *M. pneumoniae* (21%) and Chikungunya (7%). There was also evidence of reactivation of CMV (25%) and VZV (12%) infections. Serological testing found three cases of hepatitis E infection among the GBS group (5%). ZIKV infection was documented in a single GBS patient. Recent flavivirus infection was more frequent in the hospital (23%) than the house controls (4%) but frequencies of the other infections were similar. Recent *C. jejuni* infection was between 9% and 14% in the controls. Pooling together recent infection and exposed to *C. jejuni* in cases was 74% and between 45% and 56% in controls. There was no laboratory evidence of the assessed infections in 18% of GBS cases compared to 32% in hospital and 29% in house controls (Table 3). Children under 15 years old were all negative for exposure to both flavivirus and Chikungunya infections but positive for *C. jejuni* (6/7), *M. pneumoniae* (4/6) and CMV reactivation (5/7) (S3 Table). Cases of GBS with primary axonal and Miller Fisher syndrome (MFS) had a higher frequency of recent *C. jejuni* infection (40%) compared with primary demyelinating cases (23%)(OR 1·2, CI 0·20–6·97). Primary demyelinating, primary axonal and MFS variants of GBS exhibited similar

Table 2. Neurological features of GBS cases by age group and total.

| Presenting Neurological Signs and Symptoms | < 15 years old N=7 (%) | 15 – 64 years old N=37 (%) | ≥ 65 years old N=13 (%) | Total N=57 (%) |
|---|---|---|---|---|
| Days from onset to neurological symptoms, Median (Range) | 4 (1-33) | 5 (0-26) | 3 (0-35) | 5 (0–35) |
| Motor dysfunction[a] | 7 | 36 | 13 | 56 (98) |
| Ascending paralysis pattern | 3 | 23 | 7 | 33 (58) |
| Lower limb paralysis | 6 | 27 | 10 | 43 (75) |
| Upper limb paralysis | 2 | 18 | 8 | 28 (49) |
| Descending paralysis | 0 | 4 | 3 | 7 (12) |
| Sensory symptoms[b] | 2 | 21 | 8 | 31 (54) |
| Allodynia or neuropathic pain | 1 | 4 | 2 | 7 (12) |
| Facial palsy | 0 | 4 | 1 | 5 (9) |
| Dysautonomia | 0 | 4 | 1 | 5 (9) |
| Urinary incontinence | 0 | 1 | 1 | 2 (4) |
| Vertigo | 0 | 0 | 1 | 1 (2) |
| **Neurological Characteristics** | | | | |
| MRC Sum Score at nadir, Median (Range) | 24 (0-56) | 32 (0-60) | 30 (12-52) | 30 (0-60) |
| GBS disability score, Median (Range) | | | | |
| 0 – Healthy | 0 | 0 | 0 | 0 |
| 1 – Minor symptoms capable of running | 0 | 0 | 0 | 0 |
| 2 – Walks 10 m or more without assistance, unable to run | 0 | 3 | 1 | 4 (7) |
| 3 – Walks 10 m across an open space with help | 1 | 9 | 1 | 11 (19) |
| 4 – Bedridden or chairbound | 5 | 19 | 10 | 34 (60) |
| 5 – Assisted ventilation for at least part of the day | 1 | 6 | 1 | 8 (14) |
| 6 – Death | 0 | 0 | 0 | 0 |
| Modified Rankin Scale at nadir | | | | |
| 0 – No symptoms | 0 | 0 | 0 | 0 |
| 1 – No significant disability | 0 | 1 | 0 | 1 (2) |
| 2 – Slight disability | 0 | 3 | 1 | 4 (7) |
| 3 – Moderate disability | 0 | 5 | 1 | 6 (10) |
| 4 – Moderately severe disability | 4 | 9 | 5 | 18 (32) |
| 5 – Severe disability | 3 | 19 | 6 | 28 (49) |
| 6 – Death | 0 | 0 | 0 | 0 |
| ICU admission | 6 | 21/31 | 7/12 | 34/50 (68) |
| Required mechanical ventilation | 1 | 6 | 1 | 8 (14) |
| **CSF features, N=46** | N=5 | N=29 | N=12 | |
| CSF WBC count, Median (Range) | 0 (0-41) | 0 (0-22) | 0 (0-21) | 0 (0–41) |
| CSF protein, Median (Range) | 39 (25-67) | 70 (20-236) | 60 (19-283) | 67 (19–283) |
| **Neuroconduction & EMG studies[c], N=43** | N=7 | N=25 | N=11 | |
| Primary demyelinating | 5 | 8 | 5 | 18 (42) |
| Primary axonal | 1 | 10 | 4 | 15 (35) |
| Normal | 1 | 0 | 0 | 1 (2) |
| Equivocal | 0 | 4 | 0 | 4 (9) |
| Unexcitable | 0 | 2 | 0 | 2 (5) |
| Other | 0 | 1 | 2 | 3 (7) |
| **Treatment** | | | | |
| IVIG | 7 | 19 | 5 | 31 (54) |
| Plasma exchange | 0 | 14 | 6 | 20 (35) |

*(Continued)*

**Table 2.** (Continued)

| Presenting Neurological Signs and Symptoms | < 15 years old N=7 (%) | 15 – 64 years old N=37 (%) | ≥ 65 years old N=13 (%) | Total N=57 (%) |
|---|---|---|---|---|
| Steroids | 0 | 2 | 0 | 2 (4) |
| Other | 0 | 0 | 1 | 1 (2) |
| No treatment | 0 | 2 | 1 | 3 (5) |

MRC: Medical Research Council; ICU: Intensive care unit; WBC: white blood cell.

[a]Motor dysfunction is the presence of muscle weakness in the lower and/or upper limb or paralysis.

[b]Sensory symptoms are the presence of hypoesthesia, anesthesia, and/or paresthesias.

[c]Based on electrophysiological criteria by Hadden [12]

frequencies of recent *M. pneumoniae* exposure (26%, 21%, and 20%, respectively) (OR 0·76, CI 0·14–4·02). Cases of primary axonal variant showed a high frequency of CMV reactivation (60%) as compared with primary demyelinating (36%)(OR 1·07, CI 0·26–4·42) and MFS variants (20%). Similarly, cases of primary axonal had a high frequency of 2 or more co-infections (47%) as compared with primary demyelinating (29%)(OR 1·66, CI 0·26–10·37) and MFS (20%) (S4 and S5 Tables).

In the bivariate analysis, male sex, diarrhea as presenting symptom, nausea/vomiting, asthenia, and upper respiratory infection in the four weeks preceding the onset of symptoms were associated with GBS. Of these, male sex, diarrhea as presenting symptom, upper respiratory infection in the four weeks preceding the onset of symptoms, and the number of infections remained statistically significantly associated in the adjusted model (Table 4). Analysis of specific infections showed that recent/exposed infection by *C. jejuni* was statistically significant (p=0·026) in the bivariate analysis but did not enter the final multiple model.

## Anti-glycolipid profile

The profile of anti-glycolipid antibodies in GBS cases and controls is presented in Table 5. The patterns of IgG reactivity as determined by the fold-change value to anti-glycolipids tested in the GBS cases, household, and hospital controls are shown in Fig 2. IgG antibodies against the single glycolipid GM1 and the heteromeric complexes GM1:GD1a, GM1:GT1a, GM1:GQ1b, GM1:GD3, GD1:Sulfatide, GD1a:GT1a, GD1b:GT1 and GalNAc-GD1a were found to be associated with GBS in the conditional logistic model after adjustments for sex, upper respiratory infections or influenza, and number of infections. GBS cases with serological evidence of *M. pneumoniae* infection and CMV reactivation exhibited similar frequencies of IgG antibodies against the glycolipid GM1 and the heteromeric complexes observed in the GBS cases with *C. jejuni* infection (S6 Table). However, in subjects with recent/exposed infections by *C. jejuni* anti-GM1 IgG and the heteromeric complexes GM:PS, GM1:GT1a, GM1:GD3, GM1:Sulfatide, and GD1a:GT1a were associated with GBS in the conditional logistic regression models adjusted for sex, upper respiratory infection or influenza four weeks before onset or the number of infections (S7 Table). Although there was a higher frequency of anti-heteromeric complexes IgG antibodies among primary axonal cases, a relative high frequency of such antibodies was also observed in primary demyelinating cases (S8 Table).

## Discussion

The main aim of this case-control study was to assess infectious diseases as risk factors for GBS in Colombia following the 2015-2016 Zika virus epidemic when a major GBS outbreak emerged in Colombia and other Latin American countries [5,6]. There are several studies that

**Table 3. Molecular and serological diagnosis of infectious disorders in GBS cases and controls.**

| Infection assessment | GBS cases N=57 (%) | Hospital controls N=22 (%) | House controls N=55 (%) |
|---|---|---|---|
| **Zika virus RT–PCR any fluid[a], n/N** | 57/57 | 22/22 | 54/55 |
| Positive infection | 1 (2) | 0 | 0 |
| Negative | 56 (98) | 22 (100) | 54 (100) |
| Not done | 0 | 0 | 1 |
| **Flavivirus immunodiagnosis, n/N** | 56/57 | 22/22 | 52/55 |
| Recent infection | 2 (3) | 5 (23) | 2 (4) |
| Exposed | 16 (29) | 5 (23) | 22 (42) |
| Negative | 38 (68) | 12 (54) | 28 (54) |
| Not done | 1 | 0 | 3 |
| **Chikungunya immunodiagnosis, n/N** | 56/57 | 20/22 | 54/55 |
| Recent infection | 4 (7) | 1 (5) | 1 (2) |
| Exposed | 15 (27) | 7 (35) | 14 (26) |
| Negative | 37 (66) | 12 (60) | 39 (72) |
| Not done | 1 | 2 | 1 |
| ***Campylobacter jejuni* immunodiagnosis, n/N** | 57/57 | 22/22 | 55/55 |
| Recent infection | 19 (33) | 2 (9) | 8 (14) |
| Exposed | 23 (41) | 8 (36) | 23 (42) |
| Negative | 15 (26) | 12 (55) | 24 (44) |
| ***Mycoplasma pneumoniae* immunodiagnosis (IgM), n/N** | 57/57 | 22/22 | 55/55 |
| Recent infection | 12 (21) | 3 (14) | 8 (15) |
| Negative | 41 (72) | 19 (86) | 44 (80) |
| Undetermined | 4 (7) | 0 | 3 (5) |
| **Cytomegalovirus immunodiagnosis, n/N** | 56/57 | 22/22 | 55/55 |
| Primoinfection | 0 | 0 | 0 |
| Reactivation | 14 (25) | 7 (32) | 15 (27) |
| Negative | 42 (75) | 15 (68) | 40 (73) |
| Not done | 1 | 0 | 0 |
| **Epstein–Barr virus immunodiagnosis, n/N** | 57/57 | 22/22 | 55/55 |
| Primoinfection | 1 (2) | 0 | 0 |
| Reactivation | 2 (4) | 0 | 1(2) |
| Negative | 54 (94) | 22 (100) | 54 (98) |
| **Varicella Zoster virus immunodiagnosis, n/N** | 57/57 | 22/22 | 55/55 |
| Primoinfection | 0 | 0 | 0 |
| Reactivation | 7 (12) | 0 | 5 (9) |
| Negative | 50 (88) | 22 (100) | 50 (91) |
| **Hepatitis E virus immunodiagnosis (IgM), n/N** | 56/57 | 20/22 | 53/55 |
| Positive | 3 (5) | 0 | 0 |
| Negative | 53 (95) | 20 (100) | 53 (100) |
| Not done | 1 | 2 | 2 |
| **Number of infections** | N=57 | N=22 | N=55 |
| 0 | 10 (18) | 7 (32) | 16 (29) |
| 1 | 28 (49) | 7 (32) | 23 (42) |
| 2 or more | 19 (33) | 8 (36) | 16 (29) |

[a]Fluids tested included at least blood, urine and/or CSF.

**Table 4. Association analysis of clinical characteristics and infection assessment with GBS.**

| Characteristics | GBS cases N=57 (%) | Controls N=77 (%) | OR (95% CI) | p–value | aOR (95% CI) | p–value |
|---|---|---|---|---|---|---|
| **Demographic and clinical features** | | | | | | |
| Age 15 years old or older | 50 (88) | 73 (96) | 0·14 (0·01–1·16) | 0·07 | | |
| Male | 38 (67) | 18 (23) | 4·44 (2·03–9·71) | <0·0001 | 7·84 (2·49–24·70) | <0·0001 |
| Fever | 20 (35) | 24 (31) | 1·55 (0·7–3·42) | 0·26 | | |
| Rash | 3 (5) | 5 (7) | 1·09 (0·25–4·62) | 0·90 | | |
| Headache | 11 (19) | 9 (12) | 1·94 (0·69–5·46) | 0·20 | | |
| Myalgia | 16 (28) | 11 (14) | 2·66 (0·99–7·12) | 0·05 | | |
| Arthralgia | 7 (12) | 9 (12) | 1·06 (0·33–3·39) | 0·92 | | |
| Diarrhea as presenting symptom | 16 (28) | 7 (9) | 5·52 (1·57–19·37) | 0·008 | 10·94 (1·8–66·29) | 0·009 |
| Diarrhea 4 weeks before onset | 17 (30) | 1 (1) | .. | 1 | | |
| Nausea/vomiting | 11 (19) | 6 (8) | 2·94 (1–8·64) | 0·04 | | |
| Asthenia | 28 (49) | 18 (23) | 3·61 (1·6–8·12) | 0·002 | | |
| Upper respiratory infection or influenza 4 weeks before onset | 19 (33) | 5 (7) | 8·25 (2·4–28·34) | 0·001 | 13·91 (2·38–81·1) | 0·003 |
| **Infection assessment**[a] | | | | | | |
| Flaviviruses, n/N | 56/57 | 74/77 | | | | |
| Recent | 2 (4) | 7 (9) | 0·44 (0·086–2·3) | 0·33 | | |
| Exposed/negative | 54 (96) | 67 (91) | 1 | | | |
| Chikungunya virus, n/N | 56/57 | 74/77 | | | | |
| Recent | 4 (7) | 2 (2.7) | 4·97 (0·53–46·33) | 0·16 | | |
| Exposed/negative | 52 (93) | 72 (97) | 1 | | | |
| *Campylobacter jejuni* | 57/57 | 77/77 | | | | |
| Exposed/recent | 42 (74) | 41 (53) | 2·72 (1·12–6·57) | 0·02 | | |
| Negative | 15 (26) | 36 (47) | 1 | | | |
| *Mycoplasma. pneumoniae* | 53/57 | 74/77 | | | | |
| Recent | 12 (23) | 11 (15) | 1·76 (0·61–5·06) | 0·29 | | |
| Negative | 41 (77) | 63 (85) | 1 | | | |
| Cytomegalovirus | 56/57 | 77/77 | | | | |
| Reactivation | 14 (25) | 22 (29) | 0·88 (0·39–2·03) | 0·78 | | |
| Negative | 42 (75) | 55 (71) | 1 (0–6) | | | |
| Epstein Barr virus | 56/57 | 77/77 | | | | |
| Primoinfection/reactivation | 3 (5) | 1 (1) | 4·37 (0·44–43·09) | 0·20 | | |
| Negative | 53 (95) | 76 (99) | 1 | | | |
| Varicella Zoster virus | 57/57 | 77/77 | | | | |
| Reactivation | 7 (12) | 5 (7) | 2·14 (0·59–7·74) | 0·24 | | |
| Negative | 50 (88) | 72 (93) | 1 | | | |
| Number of infections | | | | | | |
| None | 10 (18) | 23 (30) | 1 | | | |
| Single | 28 (49) | 30 (39) | 2·34 (0·81–6·7) | 0·11 | | |
| Two or more | 19 (33) | 24 (31) | 2·02 (0·71–5·74) | 0·18 | | |
| Number of infections, Median (Range)[b] | 1 (0–6) | 1 (0–7) | 1·35 (0·99–1·83) | 0·05 | 1·77 (1·04–3·01) | 0·03 |

[a]Three cases of hepatitis E were diagnosed among GBS cases, but all controls were negative. The OR value is unreliable.

[b]Number of infections refers to the total number of microbiological lab tests yielding positive results.

**Table 5. Frequencies and association analyses of anti-glycolipid IgGs with GBS.**

| Anti-glycolipid IgG[a] | GBS Cases N=57 | Controls N=77 | OR (95% CI) | p–value | aOR (95% CI) | p–value |
|---|---|---|---|---|---|---|
| Any of 136 | 50 (88) | 59 (77) | 2·25 (0·85–5·98) | 0·10 | | |
| GM1 | 18 (32) | 3 (4) | 11·48 (2·63–50·11) | 0·001 | 17·4 (2·08–145·48) [b] | 0·008 |
| GM2 | 5 (9) | 3 (4) | 2·39 (0·56–10·21) | 0·24 | | |
| GT1a | 12 (21) | 2 (3) | 8·42 (1·86–38·06) | 0·006 | 5·9 (0·65–52·58) [c] | 0·11 |
| GT1b | 6 (11) | 3 (4) | 2·77 (0·68–11·3) | 0·15 | | |
| GD1a | 7 (12) | 2 (3) | 4·92 (1–24·08) | 0·04 | | |
| GD1b | 7 (12) | 2 (3) | 3·82 (0·78–15·58) | 0·09 | | |
| GD3 | 10 (18) | 6 (8) | 2·85 (0·95–8·51) | 0·06 | | |
| GalC | 11 (19) | 4 (5) | 5·78 (1·26–26·47) | 0·02 | | |
| Sulfatide | 8 (14) | 3 (4) | 7·31 (0·88–60·33) | 0·06 | | |
| GA1 | 5 (9) | 3 (4) | 1·82 (0·43–7·71) | 0·41 | | |
| GQ1b | 7 (12) | 3 (4) | 2·92 (0·74–11·5) | 0·12 | | |
| PS | 3 (4) | 4 (5) | 1·12 (0·21–5·82) | 0·89 | | |
| LM1 | 2 (4) | 2 (3) | 1·18 (0·16–8·6) | 0·86 | | |
| GalNAc–GD1a | 9 (16) | 2 (3) | 5·6 (1·19–26·28) | 0·03 | | |
| GM1:GD1a | 24 (42) | 2 (3) | 29·55 (3·97–219·92) | 0·001 | 19·2 (2·23–164·28) [b] | 0·007 |
| GM1:GT1a | 26 (46) | 2 (3) | 34·45 (4·64–255·41) | 0·001 | 51·1 (4·48–583·91) [c] | 0·002 |
| GM1:GQ1b | 27 (47) | 4 (5) | 33·5 (4·52–248·31) | 0·001 | 18·1 (2·13–154·02) [b] | 0·008 |
| GM1:GD3 | 23 (40) | 6 (8) | 7·34 (2·5–21·53) | <0·0001 | 3·9 (1·1–13·58) [b] | 0·03 |
| GM1:Sulfatide | 23 (40) | 3 (4) | 14·06 (3·28–60·21) | <0·0001 | 10 (1·95–51·24) [b] | 0·006 |
| GD1a:GT1a | 19 (33) | 3 (4) | 8·49 (2·49–28·94) | 0·001 | 7·7 (1·6–36·62) [b] | 0·01 |
| GD1b:GT1a | 17 (30) | 2 (3) | 22·42 (2·95–170·26) | 0·003 | 83·5 (3·04–2294·92) [d] | 0·009 |

[a]A total of 136 anti-glycolip IgG antibodies were analyzed.

[b]Adjusted for sex, upper respiratory infection, or influenza 4 weeks before onset and number of infections.

[c]Adjusted for sex and upper respiratory infection or influenza 4 weeks before the onset.

[d]Adjusted for sex, upper respiratory infection, or influenza 4 weeks before onset and asthenia

have focused mainly on the role of Zika infection and arboviruses as risk factors for GBS [31–34]. Our study, which covered the endemic period of Zika in Colombia between July 2016 and December 2019, reveals that 1) after the Zika virus became endemic in the summer of 2016, such viral infection is no longer the pre-eminent risk factor for GBS in Colombia, 2) clinical evidence of diarrhea as presenting symptom and preceding illnesses (within 4 weeks before onset of GBS) such as upper respiratory infection are associated with GBS, 3) rather than identifying a single specific infection statistically significantly associated with GBS, *C. jejuni* and *M. pneumoniae,* became the most frequent over several preceding infections present in GBS in the Colombian population, 4) rather than a single pathogen, the number of coinfections or infection exposures was statistically significantly associated with GBS, 5) although less frequent, other infections, such as Chikungunya, flaviviruses, and Hepatitis E, are part of a diverse group of infections associated with GBS, 6) reactivation of CMV (25%) and VZV (12%) rather than primary infection are frequent in GBS cases, and 7) the Anti-glycolipid IgG response to GM-1 and subsets of heteromeric glycolipids are potential bio-markers of GBS and, whilst a higher frequency of these biomarkers was seen among *C. jejuni* infections and primary axonal phenotypes, such pattern of antibody immunoreactivity was also present among *M. pneumoniae* and CMV positive cases as wells as the primary demyelin-ating phenotype.

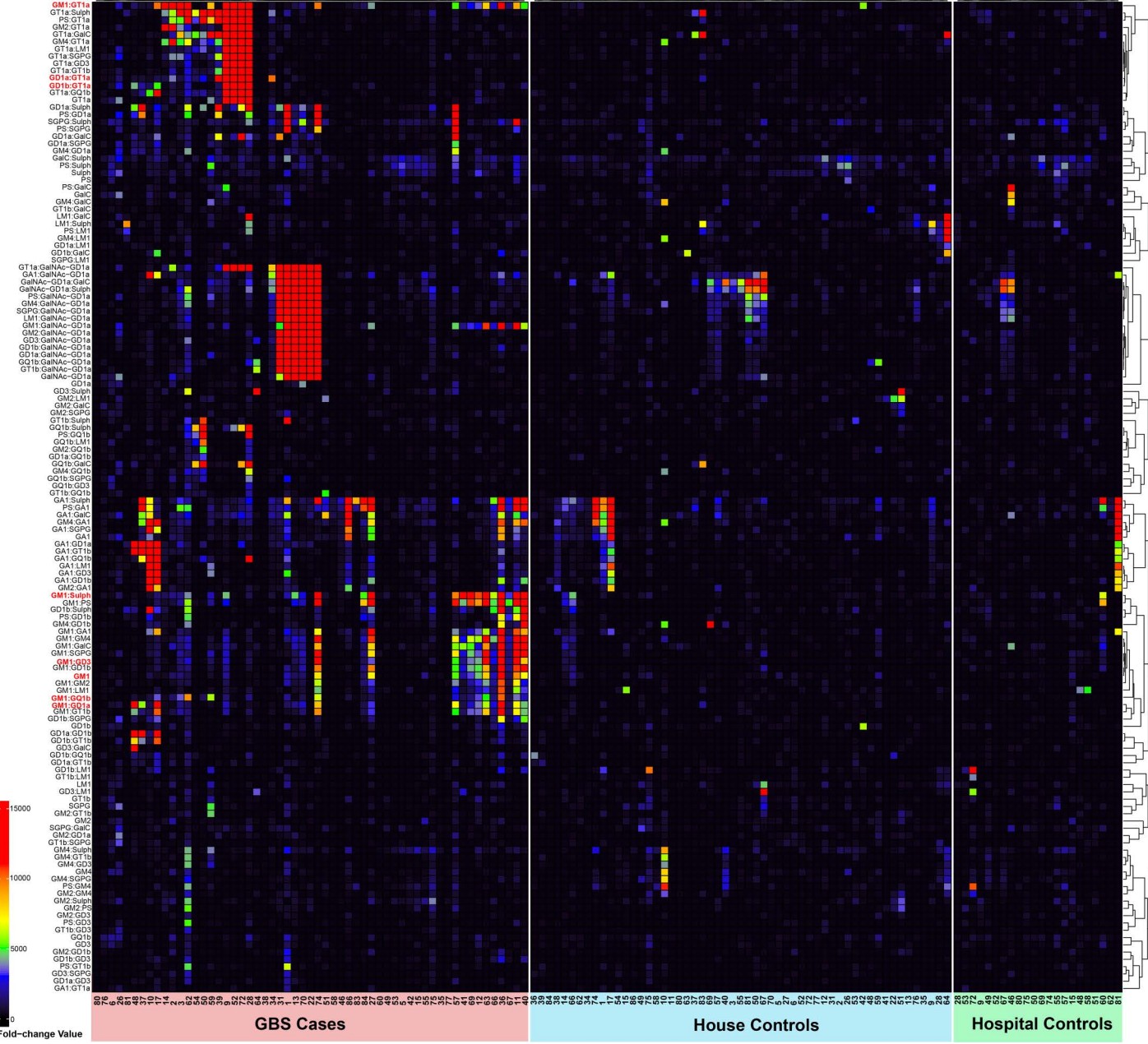

**Fig 2. Patterns of IgG reactivity to anti-glycolipids in GBS cases and controls.**

Our study underscores the role of infections in the etiopathogenesis of GBS. Similar observations have been shown in other studies [35,36] and underline the potential role of infection interactions and immunological responses to multiple infections in triggering GBS. In this study, diarrhea was one of the most common presenting non-neurological symptoms. Notably, exposure to *C. jejuni* is one of the most frequent infections in GBS cases during the post-Zika epidemic period (2016-2019) in Colombia, linking this exposure to the presence of diarrhea as one of the most important risk factors for GBS in the Colombian population. The high frequency of *C. jejuni* and diarrhea in our study concurs with other case-control studies worldwide that place *C. jejuni* infection as one of the most important risk factors for GBS in

tropical and non-tropical regions [37–39] and underscore the importance of *C. jejuni* as an etiopathogenic trigger for GBS in Latin American countries as it was revealed recently by the outbreaks of GBS in Peru in 2018-2019. [7,40]. Previous observational and case-control studies have found diarrhea to be the most frequent preceding illness in GBS [41–43] and *C. jejuni* as the most critical enteric pathogen associated with such symptoms.[36,44–50] The lack of statistical association between diarrhea and a preceding illness in our study could be explained by lack of statistical power since only one matched control reported it. The evidence that presenting diarrhea is a risk factor in GBS and infections with *C. jejuni* and hepatitis E among GBS cases also highlights the role of enteric pathogens, food-borne or water-borne, as risk factors for GBS. From the public health perspective, our results underscore the role of food safety in preventing infections that could lead to GBS. Implementing a one-health approach to *C. jejuni* and other food and water-borne pathogens may contribute to decreasing the burden of GBS worldwide.[51]

There is a wealth of epidemiological evidence highlighting the central role of *C. jejuni* to trigger autoimmune mechanisms leading to GBS. [37–39] The questions about the factors that influence the pathogenic role of such bacteria in GBS remain open. It is still uncertain whether human genetic factors or host susceptibility to develop specific autoimmune responses or specific virulent bacterial factors influence the pathogenic role of *C. jejuni* infection in GBS. Future studies should emphasize fecal sampling and microbiota studies to identify the variability of circulating *C. jejuni* such as the previously reported O:19, O:2 and O:41 strains, among others [52] as well as specific *C. jejuni* genotypes associated with GBS and, to identify other potential pathogens involved in GBS (e.g., enteroviruses [53]).

The role of respiratory infections as a significant risk factor for GBS, such as *M. pneumoniae* and likely influenza viruses, concurs with other recent case-control and observational studies worldwide [3,42,54]. However, our study was unable to establish a laboratory evaluation of other respiratory viruses that may be involved as risk factors for GBS, including influenza and enteroviruses, in which laboratory proof of recent infection or exposure is not well-established by sero-immunological assays or the molecular evidence of infection is only possible during the acute stage of respiratory illness. Regarding vector-borne infections, the relatively low frequency of infection and lack of association with GBS of arboviral infections such as Zika, dengue, or Chikungunya contrast with studies during or after the Zika outbreak in the Americas, which demonstrated the association of Zika infection with GBS [33,55], our study results indicate that arboviral infections have a relatively low impact as an etiopathogenic factor for GBS during the endemic stage of such infections. Potential explanations are the low burden of viral transmission during the endemic phase, which is influenced by the immunity already acquired by the Colombian population to those viruses or other ecological factors that affect mosquito transmission. It may also imply that the role of arboviral infections in the pathogenesis of GBS is important mainly during the phase of viral emergence or major viral outbreaks that affect populations that do not have previous exposures to such viruses as it occurred after the Zika outbreaks in French Polynesia in 2014 and Americas during 2015-2016. [4,5,34,55] or Chikungunya outbreaks in 2014. [56,57]

In contrast with previous studies that suggested a role for CMV and VZV as a risk factor for GBS [58–61], our study found evidence that only CMV and VZV reactivation was present, but no primary infection occurred. The frequency of reactivation was not significantly different among GBS cases and controls. Such a finding may suggest that rather than a primary infection by CMV or VZV as a factor triggering GBS, increased susceptibility to viral reactivation in the setting of GBS illness or a stage of relative immunosuppression that predispose to latent viral reactivation may occur during the acute phase of GBS. Alternatively, because most of the laboratory analysis of infection, recent infection, or viral reactivation (e.g., CMV

and VZV) relies on immunological assays, polyclonal B-cell activation in response to other triggering infections or the GBS-inflammatory milieu may lead to increased immunological reactivity against the tested latent herpesviruses. [62] We could not detect viral load titers to CMV in the blood of several cases of CMV reactivations as judged by the IgM positivity and IgG avidity. As the central focus of our study was to investigate infections as a risk factor for GBS, it is essential to highlight that a quarter of GBS cases did not experience any symptoms of preceding systemic illness, and 18% did not have laboratory evidence of prior infection. These observations suggest that other unknown infections or non-infectious risk factors may play roles in the etiopathogenesis of GBS. This finding concurs with other case-control studies in different regions of the world. [63,64]

The association of a particular subset of anti-glycolipid antibodies against GM1 and the heterodimers complexes GM1:GD1a, GM1:GT1a, GM1:GQ1b, GM1:GD3, GD1:Sulfatide, GD1a:GT1a, GD1b:GT1 and GalNAc-GD1a with GBS cases emphasizes the importance of such autoimmune response in the pathogenesis of GBS. Although there was an increased frequency of such antibodies in GBS cases associated with recent exposure to *C. jejuni*, a sub-analysis of the frequency of such antibodies based on pathogen exposure showed that similar antibody profiles were present in those cases exposed to *M. pneumoniae* or CMV reactivation except GM1:Sulfatide which was significantly associated with *C. jejuni* exposed GBS cases while not present in other pathogen-induced GBS. Several studies on the anti-glycolipid immune response in GBS have outlined a diversity of such antibodies and the repertoire of the anti-heterodimer complexes. While the serological detection of IgG antibodies reactive against single gangliosides (GM1, GD1a, GD1b, GT1b and GQ1b) has supported the clinical diagnosis of GBS for many decades, the heteromeric glycolipids in serum screening is not routinely performed in the diagnostic lab. The concept that heteromeric glycolipids may form neoepitopes that are uniquely recognized by IgG antibodies was first discovered in 2004 by Kaida et al[65] when they demonstrated IgG binding to a dimer complex of GD1a:GD1b in a cohort of patients, that had no or minimal binding to either glycolipid alone. Since this discovery, our studies and others have employed this concept to screen for previously unidentified immunoglobulins in peripheral neuropathy sera.[66] We have previously demonstrated the diagnostic value of inclusion of GM1:GalC as an antigen target for screening Multifocal Motor Neuropathy [67] and during an outbreak of GBS in Peru in 2019, in which patient sera was strongly associated with the presence of IgG antibodies reactive against GM1:phosphatidylserine and/or GM1:GT1a complexes, frequently with no or low reactivity to individual glycolipids.[7] It is unknown whether these glycolipid complexes form spontaneously within the lipid rafts of the plasma membrane, thereby representing true pathogenic targets in patients with peripheral neuropathy. In our experience, the presence of antibody binding to heteromeric glycolipids is more frequently associated with antibody binding to a single glycolipid target, which is optimally presented on the assay platform due to antigen spacing or glycolipid conformational alteration, which is facilitated by the inclusion of a second glycolipid, whereby the threshold of antibody detection is reduced in the assay. Although the immune response against glycolipids and molecular mimicry has been the central focus of research linking antibody-mediated mechanisms triggered by pathogens such as *C. jejuni*, other factors such as host factor susceptibility and genetically determined immune response to infections need to be explored by future studies to explain the development of GBS in populations widely exposed to enteric pathogens such as the Colombian population.

Our study has several strengths, including the standardized approach to collecting biological samples, clinical and neurophysiological assessment of GBS phenotypes, a multi-centric approach to cover different ecological regions and populations in Colombia, which may resemble other areas of Latin America, and comprehensive laboratory-based search

and standardized laboratory testing for known pathogens known to be associated with GBS. Limitations include the low enrollment of hospital-based controls, which affected the statistical power in some comparisons and limited the ability to analyze associations with the household and hospital control separately. There is selection bias in the household group due to the excess of females because it was not always feasible to randomly select these controls, and some were chosen conveniently among the patient visitors who are frequently women. Recall bias is expected as GBS cases would be motivated to recall previous history of exposures to infections or preceding symptoms more than healthy controls. The fact that *C. jejuni* was not associated with GBS in the final multiple models could be due to insufficient statistical power, because, in part, we underestimated the frequency of exposure to *C. jejuni* in the control group and, hence, the study sample size. The number of infections is likely to be underestimated because we could not ascertain the infection status of all pathogens in every sample (i.e., samples were unavailable or undetermined results). Due to the nature of the study design, we could not establish the temporal relationship between infection and onset of GBS. However, when possible, we attempted to control such limitations using immunological or virological assays to define recent exposure. In the case of *C. jejuni*, we compared the groups of recent infection and exposure using immunological assays. Still, we did not collect stool samples to document active infection by culture or PCR testing. Finally, the few pediatric cases limited our ability to analyze factors or differences in infections between children and adults.

## Supporting information

**S1 Table. GBS demographics and presenting symptoms of matched and unmatched GBS cases.**
(DOCX)

**S2 Table. Molecular and immunological testing and infectious disease diagnosis in GBS cases and controls.**
(DOCX)

**S3 Table. Microbiological test results by age group in GBS cases.**
(DOCX)

**S4 Table. Microbiological test results by GBS phenotype.**
(DOCX)

**S5 Table. Association of microbiological test results with GBS phenotype.**
(DOCX)

**S6 Table. Anti-glycolipid IgG frequency based on microbiological testing and infectious disease diagnosis.**
(DOCX)

**S7 Table. Frequencies of anti-glycolipid IgGs by study group and conditional logistic models of their association with GBS in C. jejuni positives in serum.**
(DOCX)

**S8 Table. Anti-glycolipid IgG frequency by GBS phenotype.**
(DOCX)

**S1 Appendix. Additional members of the Neuroinfections Emerging in the Americas Study (NEAS), member list.**
(DOCX)

## Acknowledgments

We are indebted to the patients and families for their participation and support provided to the study and other member of the Neuroinfections Emerging in the Americas Study (NEAS) who have supported the operations and logistics of the network.

## Author contributions

**Conceptualization:** Lyda Osorio, Beatriz Parra, Carlos A Pardo.

**Data curation:** Lyda Osorio, Beatriz Parra, David Acero-Garces, Carlos A Pardo.

**Formal analysis:** Lyda Osorio, Beatriz Parra, Susan Halstead, Hugh J Willison, Carlos A Pardo.

**Funding acquisition:** Lyda Osorio, Beatriz Parra, Hugh J Willison, Carlos A Pardo.

**Investigation:** Beatriz Parra, Martha Moyano, Reydmar Lopez-Gonzalez, Jorge A Jimenez-Arango, Jose Vargas-Manotas, Jairo Lizarazo, Gustavo E Ramos-Burbano, Mario Daniel Llanos, Fernando Rosso, Jonathan Urrego, Juan P Rojas, Christian A Rojas, Julie Benavides-Melo, Viviana A Martinez-Villota, Karina A Luque-Burgos, Adriana M Ruiz, Liliana Soto, Laura Quintero-Corzo, Jaime A Quintero, Daniela Zuluaga-Lotero, David Acero-Garces, Susana C Dominguez-Peñuela, Susan Halstead, Hugh J Willison, Carlos A Pardo.

**Methodology:** Beatriz Parra, Martha Moyano, Susan Halstead, Carlos A Pardo.

**Project administration:** Lyda Osorio, Beatriz Parra, Martha Moyano, Carlos A Pardo.

**Resources:** Beatriz Parra, Reydmar Lopez-Gonzalez, Jorge A Jimenez-Arango, Jose Vargas-Manotas, Jairo Lizarazo, Gustavo E Ramos-Burbano, Mario Daniel Llanos, Fernando Rosso, Jonathan Urrego, Juan P Rojas, Christian A Rojas, Julie Benavides-Melo, Viviana A Martinez-Villota, Karina A Luque-Burgos, Adriana M Ruiz, Liliana Soto, Laura Quintero-Corzo, Jaime A Quintero, Daniela Zuluaga-Lotero, David Acero-Garces, Susana C Dominguez-Peñuela, Susan Halstead, Hugh J Willison, Carlos A Pardo.

**Supervision:** Lyda Osorio, Beatriz Parra, Martha Moyano, Carlos A Pardo.

**Validation:** Beatriz Parra, Martha Moyano, David Acero-Garces, Susana C Dominguez-Peñuela, Carlos A Pardo.

**Writing – original draft:** Lyda Osorio, Beatriz Parra, Carlos A Pardo.

**Writing – review & editing:** Lyda Osorio, Beatriz Parra, Reydmar Lopez-Gonzalez, Jorge A Jimenez-Arango, Jose Vargas-Manotas, Jairo Lizarazo, Gustavo E Ramos-Burbano, Mario Daniel Llanos, Fernando Rosso, Jonathan Urrego, Juan P Rojas, Christian A Rojas, Julie Benavides-Melo, Viviana A Martinez-Villota, Karina A Luque-Burgos, Adriana M Ruiz, Liliana Soto, Laura Quintero-Corzo, Jaime A Quintero, Daniela Zuluaga-Lotero, David Acero-Garces, Susana C Dominguez-Peñuela, Susan Halstead, Hugh J Willison, Carlos A Pardo.

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
