## [Decision Letter · Decision Letter 0]

7 Oct 2024

Dear Prof. Pardo,

Thank you very much for submitting your manuscript "Guillain-Barré syndrome after the Zika epidemic in Colombia: a multicenter, matched case-control study" for consideration at PLOS Neglected Tropical Diseases. As with all papers reviewed by the journal, your manuscript was reviewed by members of the editorial board and by several independent reviewers. The reviewers appreciated the attention to an important topic. Based on the reviews, we are likely to accept this manuscript for publication, providing that you modify the manuscript according to the review recommendations. 

Sincerely,

Guilherme L Werneck

Section Editor

David Safronetz

Section Editor

Reviewer's Responses to Questions

**Key Review Criteria Required for Acceptance?**

**Methods**

-Are the objectives of the study clearly articulated with a clear testable hypothesis stated?

-Is the study design appropriate to address the stated objectives?

-Is the population clearly described and appropriate for the hypothesis being tested?

-Is the sample size sufficient to ensure adequate power to address the hypothesis being tested?

-Were correct statistical analysis used to support conclusions?

-Are there concerns about ethical or regulatory requirements being met?

Reviewer #1: The method section is clearly described.

Reviewer #2: The presented work aims to establish the associations or triggers of Guillain-Barré Syndrome with infections following the Zika outbreak in Colombia.

The objectives of the study are clear, with a study design appropriate for addressing the established objectives. The population is suitable for the hypothesis being tested.

Given that Guillain-Barré Syndrome (GBS) is a relatively rare disease, it is challenging to determine if the sample size is sufficient to establish associations. However, 82 cases of GBS from various healthcare centers were studied, which adds value to the sample.

The statistical methods are correct, and all ethical considerations for the research are met.

Reviewer #3: (No Response)

**Results**

-Does the analysis presented match the analysis plan?

-Are the results clearly and completely presented?

-Are the figures (Tables, Images) of sufficient quality for clarity?

Reviewer #1: The result is clearly described. There are some suggestions to perform some sub-analysis, please see the general comments section.

Reviewer #2: The results presented are clear, detailed, and structured in a way that facilitates data interpretation. The study population is thoroughly described, including the clinical, demographic, and microbiological characteristics of patients with Guillain-Barré Syndrome (GBS) and the control group. Additionally, the neurological features, cerebrospinal fluid (CSF) findings, neurophysiological studies, treatments, and assessments of prior infections in GBS cases are compared with those of the controls.

The tables provide a quantitative summary that allows for easy comparison between groups, highlighting significant differences. The bivariate and multivariate analyses are well-explained, clearly demonstrating the associations between clinical characteristics, the presence of infections, and the onset of GBS.

Furthermore, the study investigates the presence of anti-glycolipid antibodies, particularly focusing on the association of these antibodies with GBS. The results show that specific glycolipid antibodies, such as those targeting GM1 and various heteromeric complexes, are significantly associated with GBS, providing a deeper understanding of the immunological mechanisms involved.

In summary, the results are comprehensive and offer a clear understanding of the study, making it accessible to both researchers and clinical readers.

Reviewer #3: (No Response)

**Conclusions**

-Are the conclusions supported by the data presented?

-Are the limitations of analysis clearly described?

-Do the authors discuss how these data can be helpful to advance our understanding of the topic under study?

-Is public health relevance addressed?

Reviewer #1: The conclusion is clearly described

Reviewer #2: The conclusions appear to be well-supported by the data presented. The results are provided with a detailed analysis of the clinical, microbiological, and neurological characteristics of patients with Guillain-Barré Syndrome (GBS) and the controls. The data clearly show the associations between the presence of infections, anti-glycolipid antibody profiles, and the risk of developing GBS, which supports the study's conclusions. 

The limitations of the analysis are clearly described in the discussion of the study. The key points regarding the limitations include: Selection Bias, Recall Bias, Underestimation of Infections, Temporal Relationship and Pediatric Cases.

The study underscores the importance of food safety to prevent infections like C. jejuni that could trigger GBS. The discussion suggests that a approach may be beneficial in reducing the burden of GBS by preventing food- and waterborne infections.

Reviewer #3: (No Response)

**Editorial and Data Presentation Modifications?**

Reviewer #1: (No Response)

Reviewer #2: It seems clear to me, and I have no suggestions for improving the work.

Reviewer #3: (No Response)

**Summary and General Comments**

Reviewer #1: The paper has presented the association of different emerging infectious disorders with GBS in Colombia during post-Zika period. This is a very interesting and informative study. 

However, the authors are requested to address the following comments/suggestions: 

1. The objectives of the study should be clearly described in the abstract. 

2. The authors should mention whether the house controls were age-sex matched with the cases. 

3. In Table 1, % of patients with more than 15 years were presented. It would be better, if the data of different age categories could be presented. Also it would be interesting to see the infection profiles in different age groups e.g. childhood GBS, young adults or elderly and give a comparison. Authors are requested to perform a sub-analysis if possible. 

4. Is there any data of infection profile before the Zika epidemic in Colombia? If yes, it would be interesting to compare the pre and post Zika period findings in the discussion section. 

5. There are some abbreviations used inside the text. The full form of the abbreviations should be mentioned where first used.

Reviewer #2: The study provides a clear and comprehensive presentation of the data, with a detailed analysis of the clinical, microbiological, and neurological characteristics of patients with Guillain-Barré Syndrome (GBS) and controls. The use of standardized approaches for biological sample collection and clinical and neurophysiological assessment strengthens the validity of the results. The study also benefits from a multicentric approach covering various ecological regions in Colombia.

Reviewer #3: I read the paper by Osorio et al. on GBS after the Zika epidemic in Colombia. The paper is generally well-written by providing us the insight of the antecedent infection in GBS. I just have a few comments to make.

1. Why is the Modified Rankin Scale (MRS) being used and not Hughes GBS Disability Score (HGDS) for the disability assessment? Conventionally, MRS is used for stroke and HGDS is used for GBS.

2. The authors used Hadden criteria to subclassifying the electrodiagnostic subtypes, but I did not see the EDx subtypes in the results. Which EDx subtype is more commonly associated with certain pathogen? For example, axonal or demyelinating is more commonly associated with C. jejuni or CMV or etc? And what is the association of the EDx subtypes with the anti-glycolipid antibodies?

3. Diarrhea as the presenting symptoms is significant here but not diarrhea 4 weeks before onset – You mean diarrhea as the presenting symptoms of GBS? We are more interested in diarrhea as the antecedent illness, rather than the presenting symptoms here. Could you clarify?

PLOS authors have the option to publish the peer review history of their article (what does this mean? ). If published, this will include your full peer review and any attached files.

**Do you want your identity to be public for this peer review?** For information about this choice, including consent withdrawal, please see our Privacy Policy .

Reviewer #1: Yes: Nowshin Papri

Reviewer #2: Yes: Alfonso Vallejos Parás

Reviewer #3: No

Figure Files:

Data Requirements:

Reproducibility:

References

---

## [Decision Letter · Decision Letter 1]

29 Dec 2024

PNTD-D-24-00852R1

Guillain-Barré syndrome after the Zika epidemic in Colombia: a multicenter, matched case-control study

Dear Dr. Pardo,

Thank you for submitting your manuscript to PLOS Neglected Tropical Diseases. After careful consideration, we feel that it has merit but does not fully meet PLOS Neglected Tropical Diseases's publication criteria as it currently stands. Therefore, we invite you to submit a revised version of the manuscript that addresses the points raised during the review process.

Please submit your revised manuscript within 60 days Jan 28 2025 11:59PM. If you will need more time than this to complete your revisions, please reply to this message or contact the journal office at plosntds@plos.org. Please include the following items when submitting your revised manuscript:

We look forward to receiving your revised manuscript.

Kind regards,

Guilherme L Werneck

Section Editor

Guilherme Werneck

Section Editor

Shaden Kamhawi

co-Editor-in-Chief

Paul Brindley

co-Editor-in-Chief

**Additional Editor Comments :**

Please provide detailed answers to the three points raised by reviewer #3, specifically concerning (1) diarrhea as a presenting symptom or as a preceding illness/symptom, (2) whether the axonal or the demyelinating subtypes were more commonly associated with certain pathogens and anti-glycolipids and (3) the eventual fragilities of the inclusion criteria of the hospital controls.

**Journal Requirements:**

1) Please amend your detailed Financial Disclosure statement. This is published with the article. It must therefore be completed in full sentences and contain the exact wording you wish to be published.

1) State the initials, alongside each funding source, of each author to receive each grant. For example: "This work was supported by the National Institutes of Health (####### to AM; ###### to CJ) and the National Science Foundation (###### to AM).".

**Reviewers' Comments:**

Reviewer's Responses to Questions

**Key Review Criteria Required for Acceptance?**

**Methods**

-Are the objectives of the study clearly articulated with a clear testable hypothesis stated?

-Is the study design appropriate to address the stated objectives?

-Is the population clearly described and appropriate for the hypothesis being tested?

-Is the sample size sufficient to ensure adequate power to address the hypothesis being tested?

-Were correct statistical analysis used to support conclusions?

-Are there concerns about ethical or regulatory requirements being met?

Reviewer #1: The method has been described properly.

Reviewer #3: The authors have not adequately addressed my concerns in my previous comments. The main flaws of the study:

1. The authors seem do not understand the difference between diarrhea as presenting symptom or as preceding illness/symptom. This is rather confusing, as the main objective of the study is to determine the role of preceding illness in triggering GBS. It is very hard to imagine diarrhea as the presenting symptoms in GBS, as the autonomic features would not have occurred in the early phase, when it is present, it should cast doubt on the diagnosis.

2. The authors also do not seem to understand the electrodiagnostic subtype of the GBS when the Hadden criteria were used. It is imperative to know whether the axonal or the demyelinating subtypes were more commonly associated with certain pathogens and anti-glycolipids.

Lastly, the inclusion criteria of the hospital controls are also doubtful in this study. In the Methods, it was stated that patients admitted with febrile illness or suspected acute viral infection were included as hospital controls. How would one know whether this cohort of patients may develop the GBS later? The more appropriate hospital controls would be patients that were admitted for other neurological diseases (other than GBS) after the febrile illness/viral infection.

**Results**

-Does the analysis presented match the analysis plan?

-Are the results clearly and completely presented?

-Are the figures (Tables, Images) of sufficient quality for clarity?

Reviewer #1: The result has been described properly.

Reviewer #3: (No Response)

**Conclusions**

-Are the conclusions supported by the data presented?

-Are the limitations of analysis clearly described?

-Do the authors discuss how these data can be helpful to advance our understanding of the topic under study?

-Is public health relevance addressed?

Reviewer #1: The conclusion has been described properly.

Reviewer #3: (No Response)

**Editorial and Data Presentation Modifications?**

Reviewer #1: (No Response)

Reviewer #3: (No Response)

**Summary and General Comments**

Reviewer #1: The authors have addressed all the previous comments adequately.

Reviewer #3: (No Response)

PLOS authors have the option to publish the peer review history of their article (what does this mean? ). If published, this will include your full peer review and any attached files.

**Do you want your identity to be public for this peer review?** For information about this choice, including consent withdrawal, please see our Privacy Policy .

Reviewer #1: **Yes: ** Nowshin Papri

Reviewer #3: No

**Figure resubmission:**
---

## [Editor Report · Decision Letter 2]

6 Feb 2025

Dear Prof. Pardo,

We are pleased to inform you that your manuscript 'Guillain-Barré syndrome after the Zika epidemic in Colombia: a multicenter, matched case-control study' has been provisionally accepted for publication in PLOS Neglected Tropical Diseases.

Best regards,

Guilherme L Werneck

Section Editor

Shaden Kamhawi

co-Editor-in-Chief

Paul Brindley

co-Editor-in-Chief

The authors have adequately revised the content of the manuscript to meet the reviewers' requests or, when they chose not to make changes, they have provided appropriate justifications for their decision.

---

## [Editor Report · Acceptance letter]

Dear Prof. Pardo,

We are delighted to inform you that your manuscript, "Guillain-Barré syndrome after the Zika epidemic in Colombia: a multicenter, matched case-control study," has been formally accepted for publication in PLOS Neglected Tropical Diseases.

Best regards,

Shaden Kamhawi

co-Editor-in-Chief

Paul Brindley

co-Editor-in-Chief
